# Analysis of Borehole Strain Anomalies Before the 2017 Jiuzhaigou Ms7.0 Earthquake Based on Graph Neural Network

Chenyang Li [1, 2], Changfeng Qin [1, 2], Jie Zhang [1, 2], Yu Duan [1, 2], Chengquan Chi[1, 2, *]

[1] School of Information Science and Technology, Hainan Normal University, Haikou, 571158, China
[2] Key Laboratory of Data Science and Smart Education, Hainan Normal University, Ministry of Education, Haikou, China

*Correspondence to*: Chengquan Chi (575104711@qq.com)

**Abstract.** On August 8, 2017, a strong magnitude 7.0 earthquake occurred in Jiuzhaigou, Sichuan Province, China. To assess pre-earthquake anomalies, we utilized variational mode decomposition to preprocess borehole strain observation data and combined it with a graph wavenet graph neural network model to process data from multiple stations. We obtained one-
year data from four stations near the epicenter as the training dataset and data from January 1 to August 10, 2017, as the test dataset. For the prediction results of the variational mode decomposition-graph wavenet model, the anomalous days were extracted using statistical methods, and the results of anomalous day accumulation at multiple stations showed that an increase in the number of anomalous days occurred 15–32 days before the earthquake. The acceleration effect of anomalous accumulation was most obvious in the 20-day period before the earthquake, and an increase in the number of anomalous
days also occurred in the one to three days post-earthquake. We tentatively deduce that the pre-earthquake anomalies are caused by the diffusion of strain energy near the epicenter during the accumulation process, which can be used as a signal of pre-seismic anomalies, whereas the post-earthquake anomalies are caused by the frequent occurrence of aftershocks.

## 1 Introduction

Earthquakes are vibrations caused by the rapid release of energy from the Earth's crust, causing deformation. They damage
the ground, buildings, transport, and other facilities and can lead to secondary disasters such as volcanic eruptions (Nishimura, 2017), tsunamis, and epidemics, which can cause serious harm to human society and the economy. Therefore, studying earthquake precursors is crucial. Researchers have explored possible anomalies before earthquakes in many fields, including strain (Yu et al., 2021; Chi et al., 2019; Zhu et al., 2018), geomagnetism (Zhu et al., 2021; Yao et al., 2022), geothermics (Zhang and Li, 2023; Hafeez et al., 2022), subsurface fluids (Liu et al., 2014; Yadav et al., 2023), and the
ionosphere (Shi et al., 2023; Akhoondzadeh et al., 2022).

Strain, as the most direct physical quantity indicating the transition from elastic deformation to rock damage and destabilization due to stress changes, is more likely to exhibit anomalous changes in rocks before an earthquake (Yue et al., 2020). Borehole observation can capture subtle phenomena in the process of seismicity in a timely manner, and borehole strain observation data can reflect the stress and strain changes of rocks. It involves installing an instrument probe in the soil

or rock layer, tens or even hundreds of meters underground. Scholars worldwide have accumulated numerous research results on extracting and identifying pre-seismic anomalous signals using borehole strain observation data. Shu and Zhang, (1997) first used capacitive borehole strainmeters to successfully predict a magnitude 6.4 earthquake in Ush, and Kitagawa et al., (2006) observed a change in crustal strain before the earthquake in Sumatra (9.0). Chi, (2013) studied the tidal distortion strain anomalies before the Wenchuan and Lushan earthquakes and preliminarily determined that they were the

strain precursors of the two strong earthquakes. Qiu et al., (2015) analyzed significant anomalous changes in the days before the Lushan earthquake and concluded that the anomalies recorded by borehole strainmeters were related to the genesis of the Lushan earthquake.

    There are many processing methods for seismic signals, including many common and effective methods. Ma et al., (2011) used digital filtering techniques to study the body strain and barometric pressure data from Yixian station from 2002 to 2007,

removed the long-period components in the raw data, and analysed the high-frequency spectral characteristics of the body strain with the fast Fourier transform. Deng et al., (2015) used the Fourier transform to generate a spectral decomposition method for high-resolution seismic images based on the frequency-amplitude spectrum of the signal, which was applied in the extraction of weak signals from deep reflection earthquakes. Zhang, (2018) used the continuous wavelet transform method to analyse the time-frequency analysis of the borehole strain data from Guza Station, extracted the strain anomalies

in the time-frequency spectrum, and analysed the correlation between the strain anomalies and the seismic precursor anomalies. EMD method can smooth the non-smooth signals to obtain a series of components with different frequencies (IMF), by which the non-smooth, non-linear signals can be decomposed into smooth signals with different time scales (Lei et al., 2022). Yang et al., (2014) used HHT to analyse the marginal spectral features of the unexplained large tensile jumps recorded in the borehole body strain at the Qianling seismic station in February-June 2012, and judged that the main cause of

this strain anomaly was a power supply problem. However, EMD suffers from mode aliasing phenomenon, endpoint effect, and difficulty in determining the stopping condition. Compared with the recursive decomposition mode of EMD, VMD transforms the signal decomposition into a variational decomposition mode, which is essentially a set of multiple adaptive Wiener filters, and VMD can realise the adaptive segmentation of each component in the frequency domain of the signal, which can effectively overcome the mode aliasing phenomenon generated by EMD decomposition, and has a stronger noise

robustness and a weaker end-point effect than EMD. Therefore, the VMD method is suitable for analysing nonlinear nonsmooth signals such as step, jumps and burr. The VMD method has been widely used in fields such as geosciences, and the results of processing seismic signals are significantly better than the other signal processing methods mentioned above (Zhang et al., 2022; Rao et al., 2024; Liu et al., 2016; Li et al., 2018).

    Most studies on borehole strain data are limited to analyzing data from individual stations, ignoring the spatial

relationships between stations in a seismic network. Most seismic analysis methods require knowledge of the geographic locations constituting the seismic network (Van Den Ende and Ampuero, 2020). Graph neural networks (GNNs) have become a popular deep learning method with fast computing and strong feature extraction abilities. For a graph data structure composed of a seismic station network, the use of a GNN can mine additional hidden information between nodes.

For example, in the traffic flow prediction task, nodes usually represent traffic monitoring points, and node features can be divided into explicit and implicit features. Explicit features are data that can be directly observed, e.g., the speed of vehicles passing through a node, while implicit features are information indirectly obtained through model learning or data mining methods, e.g., the congestion pattern of a specific node at different times of the day is found by analysing historical and real-time data (Chen et al., 2023). At present, the application of GNNs has achieved good results in many fields, but its application in the field of earth science is still relatively small (Lin, 2022; Bilal et al., 2022; Liu, 2022). Subsequently, the method holds great potential for application in the data analysis of seismic networks.

In our case study, a method based on a variational mode decomposition-graph wavenet (VMD-GWN) model was proposed to predict borehole strain data from multiple stations, and the pre-seismic anomalies of the Jiuzhaigou earthquake were extracted based on the prediction intervals. The VMD method can automatically extract the local features of the signal, avoiding the problem of manually selecting the basis function in the traditional decomposition method and using the VMD to preprocess the measured borehole strain data. For the graphical data structure consisting of a network of seismic stations, we take the monitoring stations at different locations as nodes, and the data directly observed by each station as explicit features. By analysing the historical observation data of the stations and the distances between the stations, we can mine the implicit features such as the response patterns of different stations in different seismic events, the correlation between stations, etc. The graph wavenet model considers the characteristics of the nodes themselves as well as the spatial relationships between different nodes, uses the GWN to jointly analyze the borehole strain data from the seismic networks near the epicenter of Jiuzhaigou, and obtains the prediction results from different stations. Based on the prediction results, pre-earthquake anomalous days were extracted, the anomalous extraction results were fitted using the S-shape function, and the anomalous days were extracted and the S-shape fitting results analyzed. The rest of the paper is organized as follows: In Section II, the Jiuzhaigou earthquake is presented. In Section III, borehole data, station data, and division of the dataset are presented. Section IV presents the methodology and the model used in this study. Section V presents the parameter selection, prediction results, anomaly extraction results, and discussion. Finally, the conclusions are presented in Section VI.

**2 Case Study**

The Sichuan Basin is at the junction of the Asia-Europe Plate and the Indian Ocean Plate, and is influenced by the neighbouring mountain ranges and plateaus, forming several fracture zones, and its unique geographic location has led to frequent earthquakes within Sichuan (Zhang, 2023). On August 8, 2017, a 7.0-magnitude earthquake occurred in Sichuan, with the epicenter located in Jiuzhaigou County, Aba Prefecture, Sichuan Province, at 33.2°N and 103.82°E, with a depth of 20 km. On August 14, 2017, it was determined that 25 people were killed, 525 people were injured, 6 people were missing, and 73,671 houses were damaged (Yi et al., 2017). Over the last decade, the Jiuzhaigou earthquake was the third largest earthquake in the active tectonic zone along the eastern margin of the Ba Yan Ka La block, the first two being the Wenchuan (8.0) in 2008 and the Lushan (7.0) in 2013. Unlike the latter two, the epicenter of the Jiuzhaigou earthquake was located at

the confluence of the East Kunlun Fracture Zone of the Ba Yan Ka La block on the Tibetan Plateau, the Minjiang River Fracture, the Tazang Fracture, and the Huya Fracture (Xu et al., 2017). The topographic map of Jiuzhaigou at the epicentre is shown in Fig. 1.

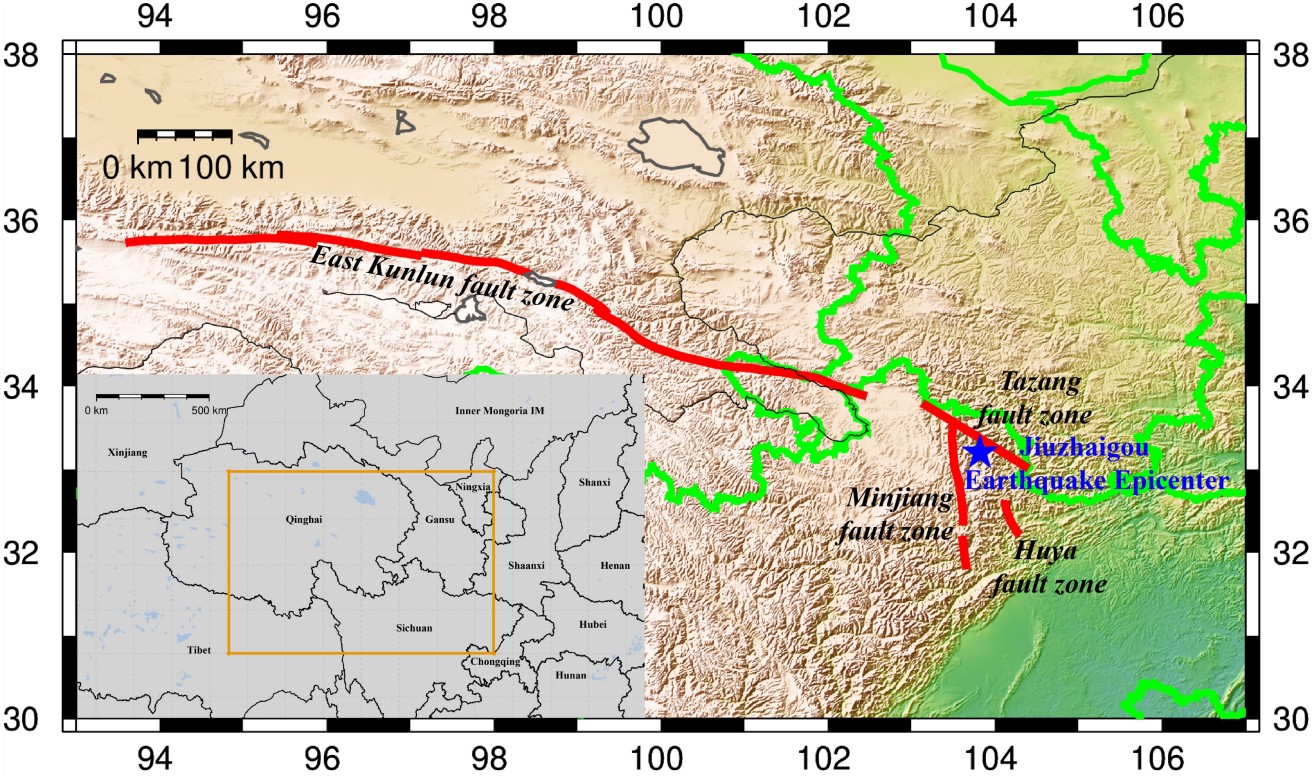

Figure 1: Topographic map of epicentre of Jiuzhaigou earthquake. Blue star indicates epicentre; red line indicates fault zone; orange boxes in grey areas indicates study locations. This map was generated by GMT software, v. 6.0.0rc5 (https://gmt-china.org/).

## 3 Data

### 3.1 Borehole strain data

Multiple recent studies have verified the reliability of sampled data from high-component borehole strainmeters, indicating that four-component borehole strain observations can detect seismic waves (Tang et al., 2023). The YRY-4 four-component borehole strainmeters has the advantages of high sensitivity, a wide observation bandwidth, self-consistent data, and long-term stability. Its working principle is to measure the relative changes between rock apertures using radial displacement sensors and record the sampled data in minutes (Zhang and Niu, 2013; Lou and Tian, 2022). Four probes mounted on the borehole strainmeters were spaced 45° apart. The measured value of any one element is recorded as $S_1$, which is rotated by

45° in turn, and the remaining three elements are recorded as $S_2$, $S_3$, and $S_4$ respectively, The amount of change in the observed values of the four elements satisfies the self-consistent equation:

$$S_1 + S_3 = k(S_2 + S_4),\qquad(1)$$

where $k$ is the coefficient that satisfies the self-consistency of the data, and we considered the data to be self-consistent when $k \geq 0.95$. The measured values of the four components were converted as follows:

$$\begin{cases} S_{13} = S_1 - S_3 \\ S_{24} = S_2 - S_4 \\ S_a = (S_1 + S_2 + S_3 + S_4)/2 \end{cases},\qquad(2)$$

All three substitutions were significant. Among them, $S_{13}$ and $S_{24}$ are mutually independent shear strains, and $S_a$ is the surface strain (Qiu et al., 2009). Compared with shear strain $S_{13}$, the surface strain $S_a$ is more representative of the four components measured by the YRY-4 borehole strainmeters, so the data characteristics of surface strain $S_a$ are used in this paper as the object of study.

**3.2 Station information**

Dobrovolsky et al. determined the relationship between the magnitude and the radius of influence (Dobrovolsky et al., 1979) using the following equation:

$$\rho = 10^{0.43M} \text{ km},\qquad(3)$$

where $M$ denotes the magnitude of the earthquake and $\rho$ denotes the radius of influence of magnitude $M$. According to the above formula, the influence range of the Jiuzhaigou earthquake is approximately 1023 km, and the data from the stations near the epicenter of the Jiuzhaigou earthquake were analyzed. Linxia station is the closest to the epicenter of the Jiuzhaigou earthquake, with a distance of 272 km, and the distance between the remaining stations and the epicenter are: Guza station at 376 km, Haiyuan station at 402 km, and Gaotai station at 775 km, indicating that these stations are capable of receiving the anomalous signals of the Jiuzhaigou earthquake. Therefore, borehole strain data from Linxia, Guza, Haiyuan, and Gaotai stations were selected as the study objects. The latitude, longitude, distance from the epicenter, and rock type of each station were analyzed, and the basic information is listed in Table I (Yang et al., 2010; Chen et al., 2024; Wu et al., 2010; Liu et al., 2016).

**Table 1.** Borehole Strain Observation Station.

| Station Name | Longitude and Latitude | Epicenter Distance(km) | Rock Type | Borehole Depth(m) | Probe Azimuth (1-way, 2-way, 3-way, 4-way) |
|---|---|---|---|---|---|
| Linxia station | 35.60◦N, 103.20◦E | 272 | Granite | 44.7 | 92、137、182、227 |
| Guza station | 30.12◦N, 102.18◦E | 376 | Proterozoic Granite | 40.69 | 52、97、142、187 |
| Haiyuan station | 36.51◦N, 105.61◦E | 402 | Mafic Rock | 36.5 | 111、156、201、246 |
| Gaotai station | 39.40◦N, 99.86◦E | 775 | Hercynian Granite | 45 | -65、-20、25、70 |

The distance between any two stations was calculated from the latitude and longitude of each station, and a distance-based matrix was constructed from the distances between the stations, which were normalized to the adjacency matrix.

Figure 2 shows the node distance map comprising the location of the epicenter of Jiuzhaigou and the distances between the stations.

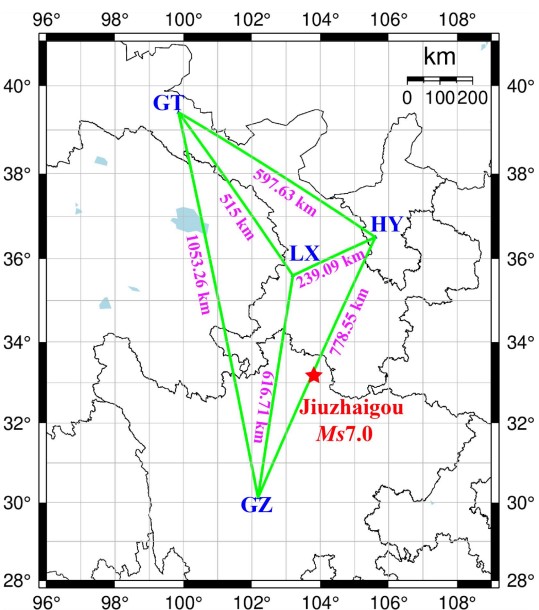

Figure 2: Nodal distance diagram constructed for borehole strain observation stations. Red star represents epicenter location. LX, GZ, HY, and GT correspond to Linxia station, Guza station, Haiyuan station, and Gaotai station, respectively. Purple represents distance between any two stations; green line represents graph structure composed of four stations.

### 3.3 Division of data set

The borehole strain data used in the study were obtained from the Beijing Seismological Bureau. First, we validated the borehole strain data from each station, and the validation results satisfied self-consistent equations. By performing strain conversion on the four-component data $S_i$ from the borehole strainmeters, two shear strains $S_{13}$, $S_{24}$ and three components of the surface strain $S_a$ were obtained, and the $S_a$ component data were used.

Through the analysis of data from each station, we identified smoother data segments to form the training dataset. Notably, the data from Gaotai and Haiyuan stations in 2013, the period between October 1, 2015, and September 30, 2016, from Guza station, and the data from Linxia station in 2014 exhibited smoother characteristics. The $S_a$ component of one year of relatively smooth borehole strain observations at the four stations was used to study the Jiuzhaigou earthquake. The first 75% of this one-year dataset was used for training, and the second 25% was used for validation, dividing the training and validation sets. As shown in Fig. 3a, the data for the $S_a$ components of the training and validation datasets are given. Since the data from Guza stations are up to August 10, 2017, the $S_a$ components of the borehole strain data from January 1 to August 10 from the four stations were selected for testing Fig. 3b. The model obtained from the training was based on relatively smooth data. Therefore, the prediction obtained from the test dataset was also relatively smooth, and the anomalies were better highlighted by comparing and analyzing the prediction results with the original data of the test set.

Figure 3 shows the raw data of the training set and the test set. Evidently, only the test set data of the Haiyuan station shows the phenomenon of a "sudden jump". Therefore, it was difficult to make an accurate judgement regarding the analysis of earthquakes based on the change patterns of raw data.

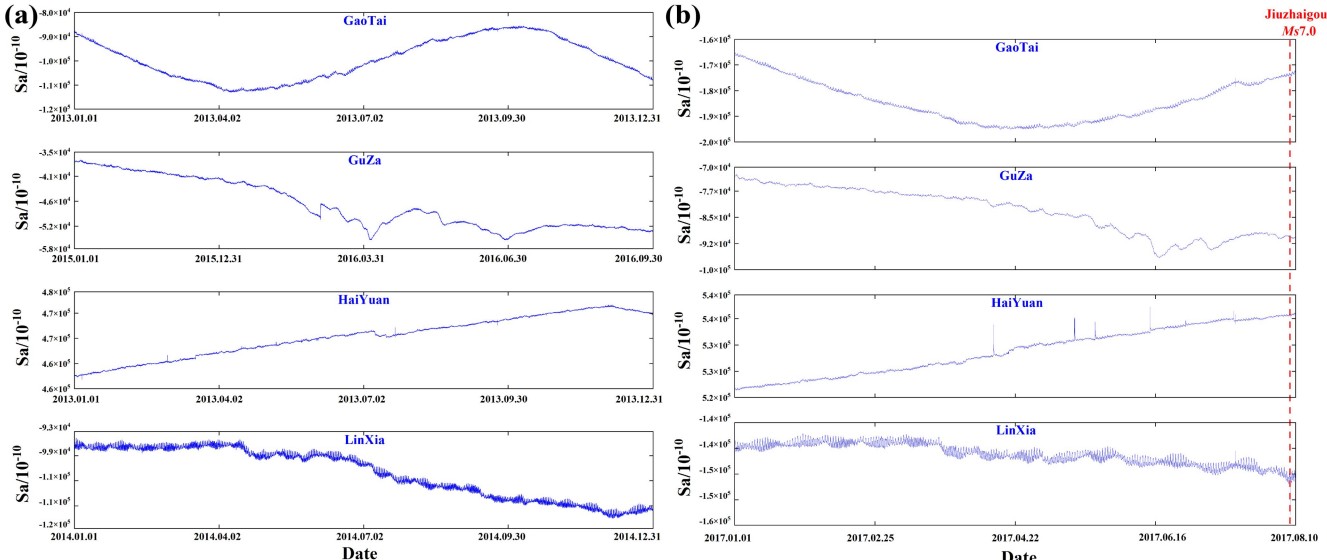

**Figure 3: Data sets of $S_a$ components for Linxia, Guza, Haiyuan, and Gaotai stations. (a) $S_a$ component data of each station for training dataset; (b) $S_a$ component data of each station for test dataset. Red dotted line indicates time of Jiuzhaigou earthquake.**

## 4 Method

### 4.1 Variational Mode Decomposition (VMD)

VMD is an adaptive, fully nonrecursive approach to modal variational and signal processing that achieves better results when dealing with nonsmooth sequence data. It achieves this by decomposing the time series data into a series of intrinsic modal functions (IMFs) with finite bandwidth and iteratively searching for the optimal solution of the variational modes (Dragomiretskiy and Zosso, 2014). The principle of the VMD algorithm is to transform the decomposition process into an optimization process, and the obtained constrained variational problem is as follows:

$$\min_{\{u_k\},\{\omega_k\}} \sum_{k=1}^{K} \left\| \partial_t \left\{ \left[ \left( \delta(t) + \frac{j}{\pi t} \right) \cdot u_k(t) \right] e^{-j\omega_k t} \right\} \right\|_2^2$$

$$s.t. \sum_{k=1}^{K} u_k(t) = f(t) ,  \tag{4}$$

where, $\{u_k\} = \{u_1, u_2, ..., u_K\}$、 $\{\omega_k\} = \{\omega_1, \omega_2, ..., \omega_K\}$ are shorthand symbols for all modes and their center frequencies, respectively. $\sum_K$ : is the sum of all modes. Solving the above equation yields the solution formula for the mode $u_K$ as:

$$\hat{u}_k^{n+1}(\omega) = \frac{\hat{f}(\omega) - \sum_{i \neq k} \hat{u}_i(\omega) + \frac{\hat{\lambda}(\omega)}{2}}{1 + 2\alpha(\omega - \omega_k)^2} ,  \tag{5}$$

the equation for solving the centre frequency is:

$$\omega_k^{n+1} = \frac{\int_0^\infty \omega |\hat{u}_k(\omega)|^2 dw}{\int_0^\infty |\hat{u}_k(\omega)|^2 dw}, \qquad (6)$$

The focus of this study was not on the VMD algorithm; it was only used for data preprocessing of the surface strain $S_a$. For a detailed explanation of the algorithm, refer to (Dragomiretskiy and Zosso, 2014).

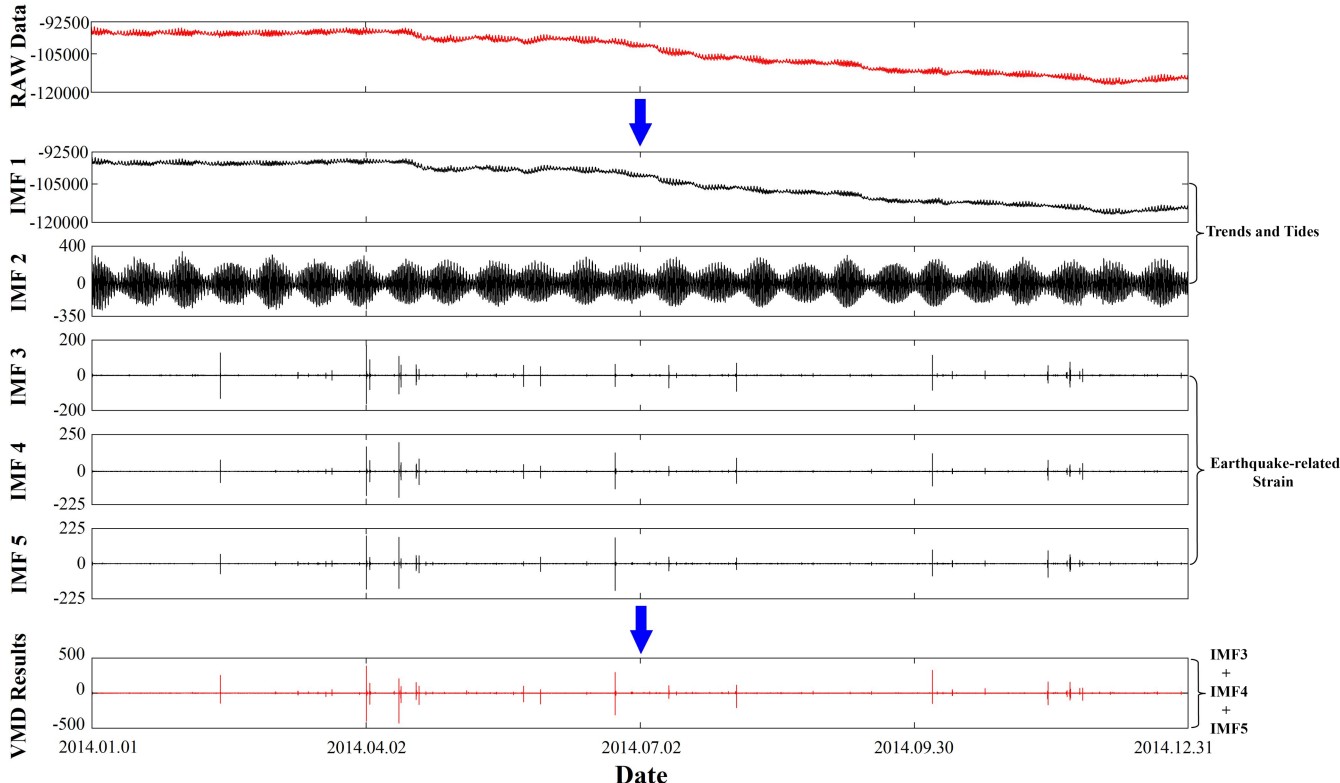

Figure 4: Plot of decomposition results of $S_a$ data using VMD method at Linxia station.

The surface strain $S_a$ data from Linxia station were selected for VMD processing. The decomposition bandwidth was set to 2000, the number of modes decomposed was five, the convergence accuracy was $10^{-7}$, and the decomposition results are shown in Fig. 4. Comparing the decomposition results with the relevant influencing factors, it was found that IMF1 represents the annual trend component and IMF2 represents the tide, removing the influence of IMF1 and IMF2 on the observed borehole strain data and summing up the remaining three components to obtain the VMD results.

### 4.2 Convolutional Neural Network (CNN)

Convolutional algorithms have been used for images; however, their applications are not limited to images. Information extraction in the time dimension through convolutional operations can process time-series data and effectively solve time-series problems. With the rapid development of deep learning and GPU arithmetic in recent years, researchers have successfully applied convolutional neural networks to time-series prediction tasks, including financial time-series prediction

(Solís et al., 2021; Kirisci and Cagcag Yolcu, 2022), wind speed series prediction (Manero et al., 2019; Gan et al., 2021), and hydrological flow forecasting (Barino et al., 2020; Shu et al., 2021).

### 4.2.1 Temporal Convolutional Networks (TCN)

In a neural network, a "layer" is a basic building block, and each layer contains a set of neurons, which accepts input data, performs specific computational operations, and then passes the results to the next layer. Different types of layers have specific functions and characteristics, and by combining and configuring different layers, powerful and flexible neural network models can be constructed to achieve a variety of complex tasks. A TCN is an improved form of CNN that has been shown to significantly outperform baseline recursive architectures in numerous sequence modeling tasks (Gopali et al., 2021;

Bai et al., 2018; Abu Bakar et al., 2021). Compared to recursive architectures, TCNs are more suitable for domains with long memories, can be processed in parallel to save time, and consist of modules such as causal and dilation convolutions. Causal convolution is a unidirectional structure that cannot detect future data and is a strict time-constrained model. Moreover, it ensures causal temporal ordering when data are extracted using feature information. When the causal convolution needs to go back for a long time for historical information, the number of convolution layers will need to be high, and problems such

as gradient vanishing will occur. Dilation convolution allows partition sampling of the convolution input. Increasing the sensory field by introducing a dilation factor enables a significant reduction in the number of convolution layers while capturing longer temporal dependencies. The relationships among the receptive field, dilation factor, and convolution kernel are as follows:

$$\begin{cases} F_n = k(n = 1) \\ F_n = F_{n-1} + (k - 1) * d(n > 1) \end{cases}'$$

(7)

where $n$ is the number of layers in the convolutional layer; $F_n$ is the sensory field of the nth convolutional layer; $k$ is the size of the convolutional kernel; and $d$ is the size of the dilation factor, which generally increases exponentially by two as the number of convolutional layers increases.

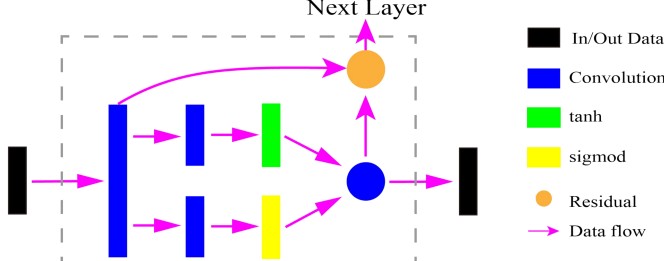

**Figure 5: Structure of gated TCN model.**

### 4.2.2 Gated TCN

Gating mechanism is an important technique in neural networks, and the core idea is to control the flow of information dynamically, so as to efficiently capture and utilise long-dependent information. The gating mechanism controls the flow of

information through the design of a "gate", which is usually a neural network layer with an activation function, whose output value is located between 0 and 1, and decides what information should be "remembered" and what information should be "forgotten" by the output value. The introduction of nonlinear gating mechanisms in sequence modeling can effectively control information transfer in hierarchical structures (Dauphin et al., 2017; Van Den Oord et al., 2016). The basic structure of the gated TCN is shown in Fig. 5; two gates were introduced after the first convolution, which are added to each convolution module of the normal TCN, one of which is used for the convolution of the input to extract features, using $tanh$ as the activation function. The other gate is used to control the processing and outflow of the information, using $sigmod$ to process it to a value between 0 and 1. The gated TCN model is expressed as:

$$T = tanh(W_1 * x + b_1) \bullet sigmod(W_2 * x + b_2) , \tag{9}$$

where $W_1$ and $W_2$ represent the weight parameters of different convolutions, $b_1$ and $b_2$ represent the corresponding bias terms, and $\bullet$ denotes the convolution operation, $T$ represent the output of the gated TCN module.

This study was based on borehole strain data, which is a typical time series, and the temporal features of the borehole strain data were extracted using a gated TCN.

## 4.3 Graph Neural Network (GNN)

Traditional deep-learning methods have achieved great success in processing Euclidean spatial data such as speech and images, whereas non-Euclidean spatial data such as social networks (Liang, 2023; Shan et al., 2024) and knowledge graphs (Li et al., 2023; Yin et al., 2024) have performed less satisfactorily when using traditional deep-learning methods. GNNs have broken new ground in many application scenarios for non-Euclidean spatial data by learning graph-structured data and extracting and mining features and patterns (Wu et al., 2020).

### 4.3.1 Graph convolution network (GCN)

The essence of a GCN is to extract the spatial features of the graph structure and achieve information transfer and feature extraction by performing a convolution operation on the feature vector through the adjacency matrix of the graph. Let the number of nodes in the graph be $N$, and the hidden state dimension of each node be $D$. The features of these nodes form a matrix of size $N * D\ X$. The relationships between individual nodes can be extracted as a relationship matrix of size $N * N\ A$. $A$ is the adjacency matrix. In the graph structure, $X$ represents the node features; $A$ represents the edge information; $X$ and $A$ are the input features of the GCN model. The convolutional layer of the graph is defined as (Kipf and Welling, 2016):

$$H^{(l+1)} = \sigma(\widetilde{D}^{-\frac{1}{2}} \widetilde{A} \widetilde{D}^{-\frac{1}{2}} H^{(l)} W^{(l)}) , \tag{9}$$

where $\widetilde{D}$ is the degree matrix of $\widetilde{A}$. $\widetilde{A}$ is an unnormalized matrix, and multiplying it directly by $H$ changes the original feature distribution. $\widetilde{A} = A + I_N$, $I_N$ is an $N$-dimensional unit matrix, and adding the unit matrix allows the node's own features to be considered. $\widetilde{D}^{-\frac{1}{2}}\widetilde{A}\widetilde{D}^{-\frac{1}{2}}$ is a symmetrically normalized matrix, $W^l$ is the weight matrix for the $l$th layer, $\sigma$ is the nonlinear

activation function, $H^{(l)} \in R^{N*D}$ is the activation matrix of layer $l$, where $H^{(0)} = X$, $H^{(L)} = Z$, and $Z$ is the final feature extracted by the GCN.

### 4.3.2 Graph wavenet (GWN)

In this section, the GWN proposed by Wu et al., (2019). The authors introduced graph convolution into the Wavenet model (Van Den Oord et al., 2016), taking into account the spatial relationships between nodes and the characteristics of the node sequence data, and employed it for a traffic prediction task. In our study, the observation stations are used as nodes, and the observation data are used as attributes of the nodes. By connecting different stations as edges and using the distance between stations as an attribute of an edge, a node graph based on multi-station borehole strain data is constituted. The node graph constructed from the distances between borehole strain stations was chosen to be constructed as an unordered graph as the information interaction between the nodes is vague. In the GWN model, each time block is followed by a graph convolution layer that extracts the temporal features of the nodes using a gated TCN and the spatial features of the nodes using a GCN.

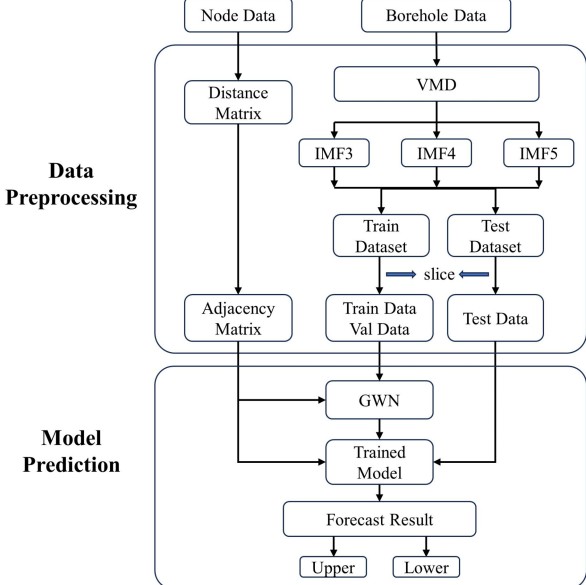

Figure 6: Flowchart of VMD-GWN model.

### 4.4 VMD-GWN model

A flowchart of the VMD-GWN model is shown in Fig. 6. It includes two parts. The first part is the data-preprocessing module. First, a distance matrix was constructed based on the distance between the nodes, and the distance matrix was normalized as the adjacency matrix. Data from each station were then collected and divided into training and test sets. Next, the training set and test set data were preprocessed separately using the VMD algorithm to obtain the required training data and test data. The second part trains the neural network and obtains the prediction results. The adjacency matrix and training

data are input into the built GWN model for training, and the adjacency matrix and test data are input into the trained model to obtain the prediction results.

## 5 Results and Discussion

### 5.1 Model hyperparameters

In this case, the VMD-GWN model was used to predict the borehole strain data at each station. The data in this study are derived from the minute observations of the borehole strainmeters; after the raw data are preprocessed using the VMD algorithm, the length of the data remains the same. According to the results of the preprocessing of the data from the four stations, a sequence of data with a size of 525600×4 was constituted as the training data set, and a sequence of data with a size of 319680×4 was constituted as the test data set. Sequence lengths of 60 and 1,440 represent the lengths of observations within an hour and within a day, respectively, and the sequence length was hourly due to equipment limitations. The model learns patterns and relationships in the data through samples and labels in the training set; the validation set is used to evaluate the performance of the model in order to adjust and optimise the model to get the best configuration and hyperparameters of the model. Our samples are the pre-processed and sliced data segments, which have a length of 60 and represent one hour of observations, and each sample contains strain data from four different stations within one hour. Our labels refer to the target values corresponding to each sample, which represent the strain data segments after one time step of the sample, and each label also contains strain data from 4 different stations within 1 hour. Slicing was done on the preprocessed training dataset based on sequence length 60. The initial 75% of the samples and labels were obtained from the training data, and the second 25% were samples and labels from the validation data. Figure 7a shows a plot of the training dataset by slicing to form the samples and labels. The same slicing process was applied to the test dataset to obtain the samples and labels. Figure 7b illustrates this slicing process for generating test dataset samples and the final shape of the predicted data. The sequence length of 60 corresponds to a one-hour prediction interval, meaning data was predicted in one-hour increments (Wu et al., 2019; Zhang and He, 2023).

**Table 2.** Optimal Hyperparameters For VWD-GWN Model.

| Hyperparameter | Learning rate | Dropout | Weight Decay rate | Training epochs | Dilation factor | Convolutional layers | Input features | Output features |
|---|---|---|---|---|---|---|---|---|
| Optimal value | 0.001 | 0.5 | 0.0001 | 100 | 1、2、4、8、16、32 | 6 | 1 | 60 |

The GWN model was divided into two modules: a 1D Gated TCN and a GCN. In this process, the convolution kernel size was set to 2, and the expansion factor grew exponentially in powers of 2. As the number of convolution layers increased and the expansion factor grew exponentially, the receptive field of each unit expanded. With the sequence length defined as 60, the number of convolutional layers was set to six. The gated TCN, incorporating dilation causal convolution, captured longer

time-series features with fewer convolutional layers. Additionally, a discard rate was applied to the GCN convolution process to control model overfitting. The optimal hyperparameters of the VMD-GWN model are listed in Table II.

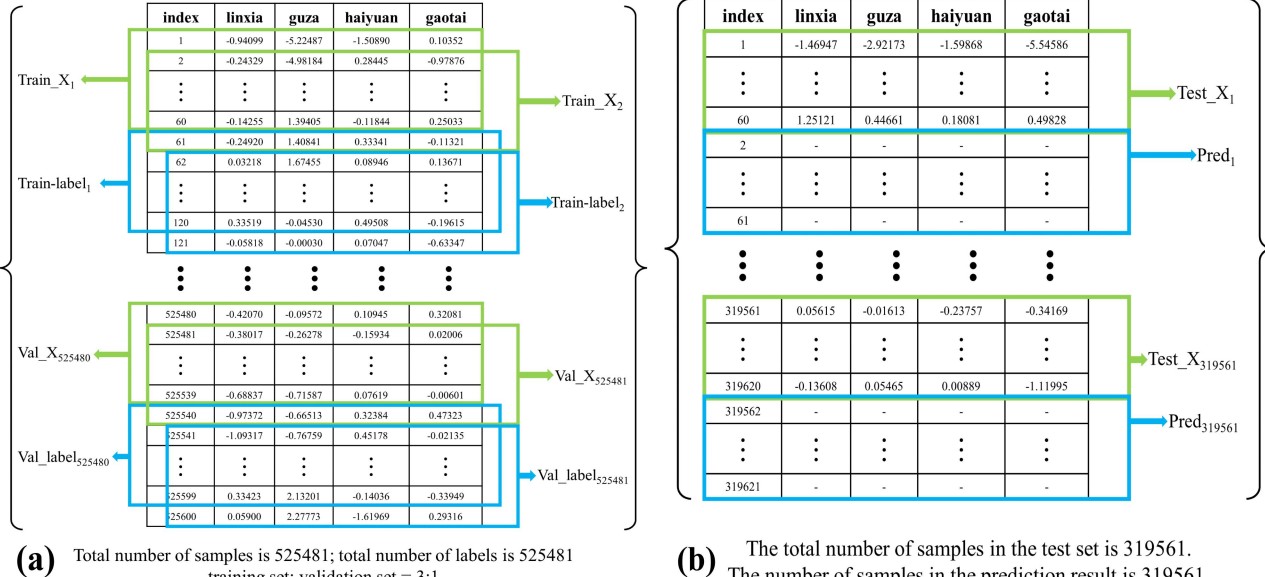

(a) Total number of samples is 525481; total number of labels is 525481
training set: validation set = 3:1

(b) The total number of samples in the test set is 319561.
The number of samples in the prediction result is 319561.

**Figure 7: Plot of data sliced to form samples and labels. (a) Data plot of samples and labels obtained from slicing the training dataset. Green box represents generated sample data; blue box represents generated label data. (b) Data graph of sample data and predicted result shapes based on test dataset slices. Green box represents sample data; blue box represents predicted result shape.**

## 5.2 Prediction results

The training data and neighbor matrix were used after preprocessing to train the GWN neural network model, and the validation data and neighbor matrix were used to evaluate the trained model, select the optimal model based on the evaluation results, and place the test data and neighbor matrix into the optimal model to perform the prediction. According to the prediction results, the data of each time step were compared; the maximum value of each time step was taken as the upper bound of the prediction interval, the minimum value was taken as the lower bound of the prediction interval, and the prediction interval was constructed according to the upper and lower bounds. Raw data and prediction intervals were analyzed to determine abnormal results. As shown in Fig. 8, the details of the prediction intervals and raw data of the Linxia station are given.

As shown in Fig. 8a, the prediction results are relatively smooth at certain peak and trough positions. In Fig. 8b, the prediction results are provided in detail and show that the prediction intervals exhibit a similar trend to the true values, particularly for some peak and valley values of the true data. In Fig. 8c, most of the true values were wrapped within the prediction intervals, whereas the values outside the prediction intervals were defined as anomalies by defining their corresponding points as anomalies.

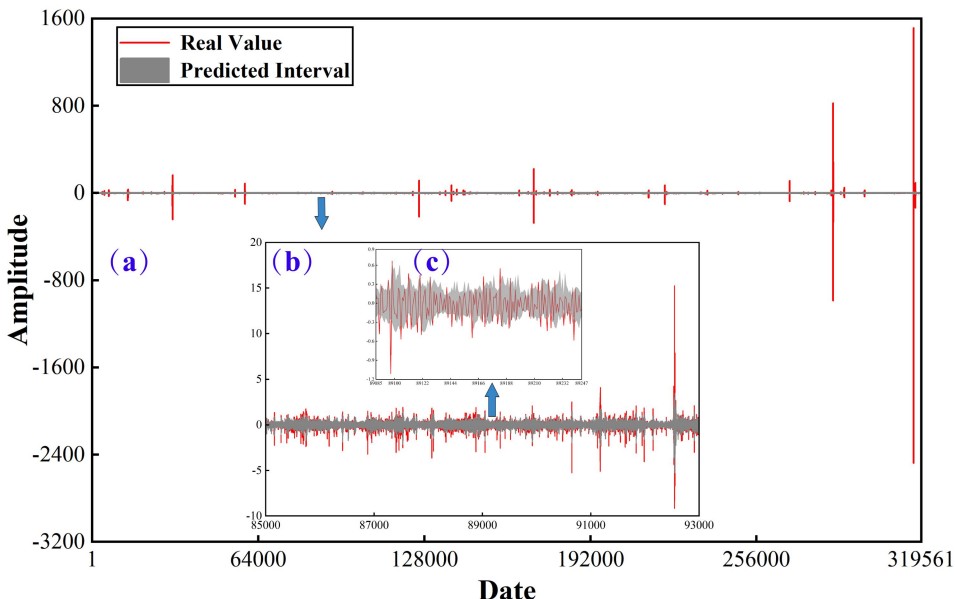

**Figure 8: Detailed plot of prediction results of VMD-GWN model at Linxia station with raw data. Red line is raw data after preprocessing and grey area is prediction interval. (a) Comparison plots of raw data and prediction intervals at Linxia station. (b–c) Detailed plots of raw data versus predicted intervals.**

## 5.3 Abnormal Extraction

For the extracted anomalies, it is difficult to judge the anomalous days before and after the onset of the earthquake; thus, we provide the judgment criteria for the anomalous days: (a) there must be more than 15 points outside the interval in a 30-minute period; (b) the difference between the centroid of the predicted interval and the actual value must be greater than 1.5 times the bandwidth of the interval, and there must be more than three such points in that 30-minute period. Days that satisfied the above conditions were defined as anomalous days (Chi et al., 2023).

Bufe and Varnes, (1993) and Bufe et al., (1994) found that the clustering of intermediate events prior to a large shock leads to a regional increase in the cumulative Benioff strain $\varepsilon(t)$, which can be fitted by a power-law time-destruction relationship:

$$\varepsilon(t) = A + B(t_f - t)^m \tag{10}$$

where $A$ and $B$ are constants, $0 < m < 1$ is a constant for adjusting the power law, $t_f$ is the predicted time of the mainshock, i.e., the critical point in time for the acceleration process of the cumulative Benioff strain (cumulative energy). This behaviour has been interpreted as a critical process preceding the movement of a large earthquake towards a critical point (i.e., the mainshock). Bufe and Varnes, (1993) justify Equation 10 with a simple model of damage mechanics. De Santis, (2014) studied the 2009 L'Aquila and 2012 Emilia earthquakes based on seismic catalogues, showing concretely how this accumulation of energy in space and time manifestations. The research idea of this paper is the extraction of multi-station pre-earthquake anomalies based on spatio-temporal features, and the fitting method proposed above has good results in spatio-temporal and this fitting method has theoretical support and physical significance, so for the anomalous results in our

original manuscript, we use the S-type function to do the fitting. De Santis et al., (2017) used Swarm magnetosatellite data to study the 2015 Nepal earthquake and proposed an S-shaped fitting function in anomalous cumulative analysis; they found that S-shaped fitting was significantly superior to linear fitting. In this case, we studied the number of anomalous day accumulations two months before and three days after the earthquake and fitted the number of anomalous day accumulations with the S-shape function, and the results of the S-shape function fitting for the four stations are shown in Fig. 9. We

constructed the horizontal coordinates centered on the date of the Jiuzhaigou earthquake, where the negative sign indicates the number of days before the earthquake and the number of days after the earthquake without a negative sign, and EQ indicates the date of the earthquake (August 8, 2017) in the center.

Through an analysis of the results of the accumulation of anomalous days two months before and three days after the earthquake, we reached the following conclusions:

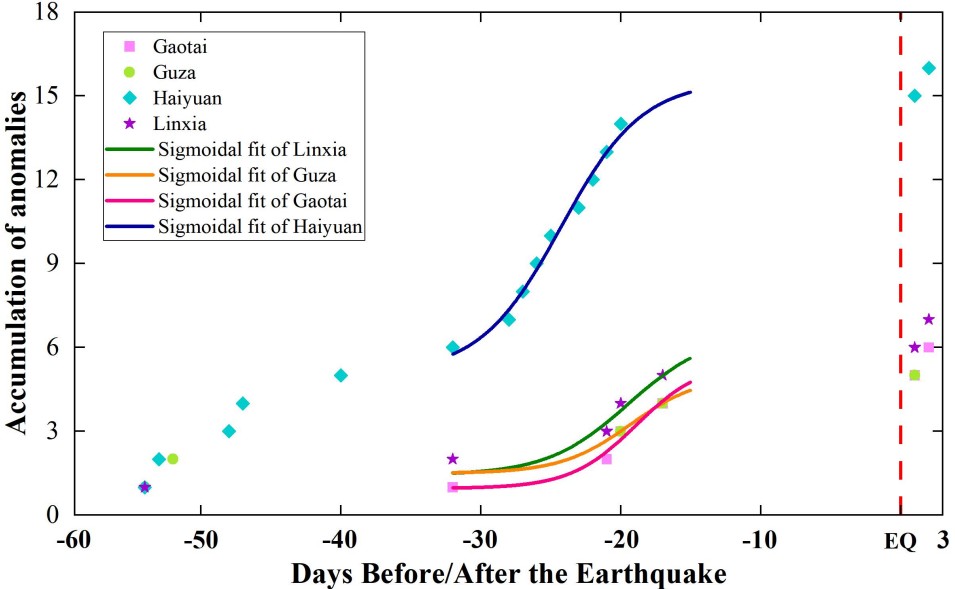

**Figure 9: Fitting results of cumulative number of anomalous days for four stations. Red dotted line represents time of earthquake; different types of dots indicate anomalous days at stations; curves of different colours represent results of S-shaped fit of anomalous accumulation of stations.**

The number of anomalous days began to gradually increase from July 7, and after continuing for 14 days until July 24 (15

350 days before the earthquake), a brief 15-day pre-earthquake quiet period began to occur. Zhong et al., (2020) studied thermal infrared (TIR) data before the Jiuzhaigou earthquake and found that there was a significant increase in TIR anomalies from July 3 to 24, which coincided almost exactly with our study period. Notably, we found that for all four stations in our study, the number of anomalous days increased simultaneously, and the S-function fitting results showed significantly accelerated anomalies. In order to verify the possibility that we extracted anomalies at all of our multiple stations, we analyzed the

355 results of other studies using other data. Xu and Li, (2020) used seismic observation records from 31 stations and found that the peaks occurred from July 11 to 21 prior to the earthquake and that the high values at the 31 stations were congruent. Xu

et al., (2024) used broadband seismometer data from ten stations near the epicenter to calculate the alignment entropy of the ground motion velocity and found that the entropy decreases were observed at all stations from July 14 to 22. Therefore, we believe that the simultaneous occurrences of anomalies at the selected stations were not coincidental. For abnormal days, we only judge whether it is abnormal, and do not know the specific number of anomalies. Therefore, we made a count of the judgement results that met the conditions in each abnormal day, and took the statistical result as the number of abnormalities per day, and calculated the abnormal rate per day based on the number of abnormalities per day, and the statistical result of the abnormal rate per day is shown in Fig. 10 below.

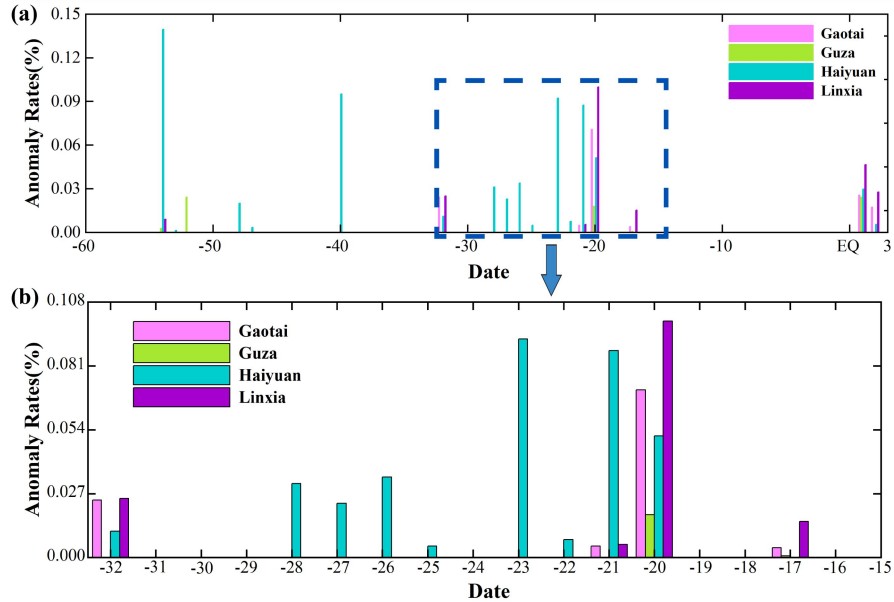

Figure 10: Daily anomaly rate statistics for four stations. Different coloured bars represent the daily anomaly rates of different stations (a) Daily anomaly rate statistics from 62 days before to 3 days after the earthquake. (b) Daily anomaly rate statistics from 32 days before the earthquake to 15 days before the earthquake.

As shown in Fig. 10(a), in the time range from 60 days before the earthquake to 33 days before the earthquake, all four stations showed only a very small number of anomalies until 32 days before the earthquake, when anomalies appeared at a number of stations, and the anomalies also increased significantly, with Haiyuan station showing the most significant number of anomalies. The dashed box in Fig. 10(a) corresponds to the time period from 32 days before the earthquake to 15 days before the earthquake, and the details are shown in Fig. 10(b). We find that there are several stations with anomalies at the same time in the 32, 21, 20, and 17 days before the earthquake, among which all four stations have anomalies on the 20th day before the earthquake, and from the value of the anomaly rate, the anomaly rate on the 17th day before the earthquake has a very obvious decrease, so we believe that a turning point occurs on the 20th day before the earthquake, which corresponds to the time when the accelerating effect of the S-shape fitting is the most obvious. After the 17th day before the earthquake, there was a quiet period where no station detected anomalies until the earthquake occurred. The combined analysis of Fig. 9 and Fig. 10 gives a fuller picture of the process of pre-seismic anomalous changes.

Zhang et al., (2018) analysed the temporal and spatial evolution characteristics of the precursor anomalies of the Jiuzhaigou earthquake, and found that the short-term phases of the precursor anomalies of the Jiuzhaigou earthquake are divided into two phases, γ1 and γ2, among which the anomalies in the γ2 phase are deformation anomalies, which are manifested as the expansion of anomalies from the near-source area to the outside of the epicentre. Wang et al., (1984) found that the extension of precursor anomalies to the periphery of the epicentre was due to the subcritical extension of cracks, and justified their conclusions based on the inversion results of the precursor observations of resistivity. Guo et al., (2020) analysed the deformation process in the unstable state of a fault and defined the meta-stable (or sub-stable) state of a fault as the transition stage from peak stress to fast destabilising critical stress throughout the slow loading and fast unloading. At this stage, the accumulated strain energy starts to be released. Based on the conclusion of Yu et al., (2019), we believe that the anomalies extracted from multiple stations at the same time were in the final stage of the formation of the Jiuzhaigou earthquake, starting in July 2017, where the anomalies in the epicenter area gradually decreased and began to diffuse to the periphery, and the stresses in the diffusion area of the epicenter were in the stage of cumulative enhancement. The anomalies received from our four stations came from the accumulation of strain energy during the diffusion process and eventually due to the accumulation of strain energy exceeding the medium-strength limit, leading to the occurrence of the Jiuzhaigou earthquake.

## 6 Conclusion

In this study, employed a VMD-GWN method to study the anomalies of multistation borehole strain data prior to the Jiuzhaigou earthquake. The influence of annual trends and tides on the borehole observation data was removed using VMD. The GWN network predicted the smooth data from each station in 2017, and anomalies were extracted by comparing the prediction results with the original data. Anomalous days were defined, and their cumulative results were fitted using an S-shaped function. Analysis showed that 15 to 32 days before the earthquake, the number of anomalous days increased at all stations, with the most significant acceleration observed in the 20 days prior to the earthquake. An increase in anomalous days was also noted one to three days after the earthquake. We believe the pre-earthquake anomalies are due to the diffusion of strain energy in the epicentral region, indicating proseismic anomalies, while post-earthquake anomalies result from aftershocks. Given the complexity and variability of earthquakes, further research is needed to refine the extraction and identification of pre-seismic borehole strain anomalies.

*Data availability.* The data that support the findings of this study are available from the China Earthquake Networks Center, but restrictions apply to the availability of these data, which were used under license for the current study, so are not publicly available. Data are however available from the Corresponding author (Email: 575104711@qq.com) upon reasonable request and with permission of the China Earthquake Networks Center.

*Author Contributions.* Conceptualization, Chenyang Li and Chengquan Chi; Data curation, Chenyang Li, Chengquan Chi and Changfeng Qin; Formal analysis, Chenyang Li, Chengquan Chi, and Jie Zhang; Investigation, Chenyang Li, Chengquan Chi and Yu Duan; Methodology, Chengquan Chi; Resources, Chengquan Chi; Software, Chenyang Li and Changfeng Qin; Supervision, Chengquan Chi; Validation, Chenyang Li, Chengquan Chi; Writing – original draft, Chengquan Chi and Chenyang Li; Writing – review & editing, Chenyang Li and Chengquan Chi.

*Competing interests.* The authors declare that they have no conflict of interest.

*Acknowledgments.* The authors would like to thank Qiu Z. H., Tang L., and Yang D. H. from the China Earthquake Administration for giving essential help in accessing the website and downloading the strain data. The authors are also grateful to the China Earthquake Networks Center for borehole strain data.

*Financial support.* This work was supported by the Hainan Provincial Natural Science Foundation of China under Grants 622RC669.

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
