# Peer review of "Analysis of Borehole Strain Anomalies Before the 2017 Jiuzhaigou Ms7.0 Earthquake Based on Graph Neural Network"

_EGUsphere, 2024_

## Author Comment (AC1)

**Response to Reviewer:**

I am very grateful to your comments for the manuscript. Thank you for your advice. All your suggestions are very important. They have important guiding significance for our paper and our research work. We have revised the manuscript according to your comments. The response to each revision is listed as following:

**General Comment**

The manuscript entitled "*Analysis of Borehole Strain Anomalies Before the 2017 Jiuzhaigou Ms7.0 Earthquake Based on Graph Neural Network*" presents the results of analyzing strain meter data from four sites prior to a large magnitude earthquake aiming to identify pre-seismic signal using Neural network technique.

My general feeling reading the manuscript is that is well organized, interesting work providing some new useful information of exploiting graph neural network approach to estimate/define pre-seismic signal on strain time series. I think that the manuscript require some small modifications and some of its parts to be improved in order to be more explanatory and understandable but in any case, it is considered, in my opinion, as a nice work. However, not being very familiar with the neural networks I would like to see some more detail information concerning the analysis in some of the manuscript paragraphs.

As a first remark I would like to mention that although I am not a native English person, in several cases the text has to be reformatted, to be "easier" for the reader.

Another general comment that I have is that the text and the processing focus on the pre-seismic period and ignores completely the post seismic period. Of course, it is well explained by the authors, that the manuscript examines the possible anomalies on the preparatory stage of a strong earthquake, but since the earthquake occurred in 2017, and a long time passed since this event, it would be interesting to see if the area, and the strain data from the same stations, shows any similar behavior as during the pre-seismic period. In other words, as a validation of this processing it could be interesting to examine the years after the earthquake if there was another period that these 4 stations show anomalous days (as in Figure 9) without the occurrence of an earthquake. The authors used data only 1 year before the earthquake … using longer period is it possible that there is and another "anomalous" "S" shape period? Of course I can understand that the authors focus on the technique as a tool for extracting pre-seismic signals of an earthquake, but I would like to see some comment on this issue.

**Response**:

Thanks for your suggestion.

(1) Thank you for pointing out that the text may need reformatting in certain areas. We understand the importance of clear and accessible language for the reader. We will carefully review and revise the manuscript to ensure that the content is more fluid and easier to understand. Additionally, we will seek the assistance of a native English

speaker to help polish the language and improve the overall readability and accuracy of the text.

(2) First of all, I apologize for the limitation that our data only extends up to August 2017, making it impossible to verify whether similar anomalies occurred in the years following the earthquake. However, reviewers can refer to our previously published article, *Pre-earthquake Anomaly Extraction from Borehole Strain Data Based on Machine Learning* (Chi et al., 2023). In that study, we analyzed data from the Guza station (one of the stations studied in this manuscript) in relation to the 2008 Wenchuan Ms 8.0 earthquake and the 2013 Lushan Ms 7.0 earthquake. We found that a distinct "anomalous" S-shaped period appeared before and after both of these large earthquakes, as shown in Figure 1A. To further investigate whether similar "anomalous" S-shaped periods occurred during other timeframes, we conducted a cumulative analysis of anomalies from 2009 to 2011. The results, shown in Figure 1B, indicate that no "anomalous" S-shaped period was observed during this time.

[Figure]

**Figure 1: Accumulated results of the abnormal days of borehole strain data at Guza station from 2007 to 2013 (Chi et al., 2019).**

Based on the above analysis, we believe that the "anomalous" S-shaped period identified in the manuscript before the earthquake is reasonable. Furthermore, the method we employed demonstrates its effectiveness as a tool for extracting pre-seismic anomaly signals.

**Some more detail comments:**
**Comment 1**
Line 16 "pro-seismic", I think that is more common the use of the term "pre-seismic".
**Response**:
  Thanks for your suggestion.
Modified "pro-seismic". It is modified to "pre-seismic".

**Comment 2**
Line 19 "…such as volcanic eruptions…" I am not so sure that earthquakes trigger

volcanic eruptions, if so please add a reference.

**Response**:

Thanks for your suggestion.

Modified "…such as volcanic eruptions…". It is modified to "…such as volcanic eruptions (Nishimura, 2017)…".

Nishimura used the data of large earthquakes and volcanic eruptions with a global magnitude of 7.5 or above as the research object, and analyzed the cumulative quantitative changes of volcanic eruptions at different distances within 5 years before and after these large earthquakes. Their results show that within 5 years after a major earthquake, the probability of volcanic eruptions within 200 km from the epicenter increases by about 50 % (Nishimura, 2017).

**Comment 3**

Line 41 "…the use of a GNN can mine additional hidden information between nodes…" this is a very general statement, please provide some examples.

**Response**:

Thanks for your suggestion.

In order to better understand "using GNN to mine additional hidden information between nodes", we give an example of a traffic flow prediction task.

Add this example to line 43 of the original manuscript: For example, in the traffic flow prediction task, nodes usually represent traffic monitoring points, and node features can be divided into explicit and implicit features. Explicit features are data that can be directly observed, e.g., the speed of vehicles passing through a node, while implicit features are information indirectly obtained through model learning or data mining methods, e.g., the congestion pattern of a specific node at different times of the day is found by analysing historical and real-time data (Chen et al., 2023).

**Comment 4**

Lines 49-51. Please describe a bit more analytical the meaning of a "node" how a node is defined? What is its characteristics and/or its physical meaning.

**Response**:

Thanks for your suggestion.

A definition of "node" has been added to line 49 of the original manuscript: For the graphical data structure consisting of a network of seismic stations, we take the monitoring stations at different locations as nodes, and the data directly observed by each station as explicit features. By analysing the historical observation data of the stations and the distances between the stations, we can mine the implicit features such as the response patterns of different stations in different seismic events, the correlation between stations, etc.

**Comment 5**

Figure 1. It would be nice to be added a map of the broad area (as inset) indicating the position of your area so to be easier for readers not familiar with the area to orientate themselves.

**Response**:

Thanks for your suggestion.

We have modified Figure 1 in the original manuscript by adding a map of the broad area (as inset) and labelling the location of the study area.

[Figure]

Figure 2: Topographic map of epicentre of Jiuzhaigou earthquake. Blue star indicates epicentre; red line indicates fault zone. This map was generated by GMT software, v. 6.0.0rc5 (https://gmt-china.org/).

**Comment 6**

Line 146 ".. preprocessing of the surface strain $S_a$" Why the authors choose only the $S_a$ and not $S_{13}$ or $S_{24}$? Please explain. A comment on this issue (selection) could be added maybe in the end of Section 3.1 line 88.

**Response**:

Thanks for your suggestion.

In order to verify that our results are also applicable to $S_{13}$ or $S_{24}$, we selected the shear strain $S_{24}$ data from four stations, namely, Guza, Xiaomiao, Luzhou and Zhaotong, and analysed them using the same method. The anomalous day accumulation results of shear strain $S_{24}$ data from the four stations are shown in Fig. 3.

[Figure]

**Figure 3: Fitting results for the accumulation of anomalous days of S₂₄ component at four stations. Red dotted line represents time of earthquake; different types of dots indicate anomalous days at stations; curves of different colours represent results of S-shaped fit of anomalous accumulation of stations.**

[Figure]

**Figure 4: Fitting results for the accumulation of anomalous days of Sₐ component at four stations. Red dotted line represents time of earthquake; different types of dots indicate anomalous days at stations; curves of different colours represent results of S-shaped fit of anomalous accumulation of stations.**

We have processed the data of shear strain $S_{24}$ by using the same method and the results are shown in Fig. 3. The processing results of the surface strain $S_a$ are given in Fig. 4. We compared Fig. 3 with Fig. 4 and found that the processing results of shear strain $S_{24}$ and surface strain $S_a$ were very similar, and both showed similar acceleration scenarios at the same time period before the earthquake.

Qiu et al analyzed the borehole strain data of Guza station before Lushan Ms7.0 earthquake. It was found that due to the large fluctuation of the Dadu River flow, the borehole strain observation curve also showed reverse large fluctuation synchronously. However, for the observation components in different directions, the fluctuation amplitude of the borehole strain observation curve is quite different, so the shear strain converted by strain may not be able to fully receive the abnormal signal of all components (Qiu et al., 2015). According to Eq. (2) in the manuscript, compared with the shear strain $S_{24}$, the surface strain $S_a$ is more representative of the four components measured by the YRY-4 borehole strain gauge, and we believe that the surface strain $S_a$ represents the sum of the anomalous results of the four components, so we use the characteristics of the data of the surface strain $S_a$ as the object of study in this paper.

An explanation was added at the end of Section 3.1, line 88, of the original manuscript: Compared with shear strain $S_{13}$, the surface strain $S_a$ is more representative of the four components measured by the YRY-4 borehole strain gauge, so the data characteristics of surface strain $S_a$ are used in this paper as the object of study.

**Comment 7**

Figure 4. It would be helpful the figure caption to be more analytic. Especially as what it is presented in the final diagram.

**Response**:

  Thanks for your suggestion.

We have revised Figure 4 in the original manuscript to provide a more detailed description of the information in Figure 4.

[Figure]

**Figure 5: Plot of decomposition results of $S_a$ data using VMD method at Linxia station.**

**Comment 8**

Sections 4.2.1 and 4.25.2 I would like (personally) to see some more detail description of this part. Actually, I would like to be more explicable and defines (maybe with examples) some of the terms used on it … "layers", "gating mechanisms" etc.

**Response**:

  Thanks for your suggestion.

Added a more detailed explanation and definition of "layer" in line 163 of the original manuscript: In a neural network, a "layer" is a basic building block, and each layer contains a set of neurons, which accepts input data, performs specific computational operations, and then passes the results to the next layer. Different types of layers have specific functions and characteristics, and by combining and configuring different layers, powerful and flexible neural network models can be constructed to achieve a variety of complex tasks.

Added a more detailed explanation and definition of "gating mechanism" in line 180 of the original manuscript: Gating mechanism is an important technique in neural networks, and the core idea is to control the flow of information dynamically, so as to efficiently capture and utilise long-dependent information. The gating mechanism controls the flow of information through the design of a "gate", which is usually a neural network layer with an activation function, whose output value is located between 0 and

1, and decides what information should be "remembered" and what information should be " forgotten " by the output value.

**Comment 9**
Equation 9. Define the parameter "T"
**Response**:
        Thanks for your suggestion.
The parameter "T" is the output of the gated TCN module, and we have added the definition of the parameter "T" to line 187 of the original manuscript.

**Comment 10**
Line 237 "…75% of the samples and labels." Please define what are the samples and what are the labels.
**Response**:
        Thanks for your suggestion.
A more detailed explanation and definition of "samples and labels" has been added to line 236 of the original manuscript: the model learns patterns and relationships in the data through samples and labels in the training set; the validation set is used to evaluate the performance of the model in order to adjust and optimise the model to get the best configuration and hyperparameters of the model. Our samples are the pre-processed and sliced data segments, which have a length of 60 and represent one hour of observations, and each sample contains strain data from four different stations within one hour. Our labels refer to the target values corresponding to each sample, which represent the strain data segments after one time step of the sample, and each label also contains strain data from 4 different stations within 1 hour.

**Comment 11**
Section 5.2 and Figure 9. Concerning the results presented in this section, could the authors comment on why the Haiyuan station shows this strong S shape anomalies, with many points, while the other stations (Linxia and Guza), although it appear that there are closer to the epicenter do not reveal a similar "strong" anomaly.
**Response**:
        Thanks for your suggestion.
We have also noticed this phenomenon you mentioned. In earthquake precursor studies, the reasons for the phenomenon that stations closer to the epicentre receive fewer anomalous signals than stations farther away may be:
(1) Differences in geological structures: The propagation characteristics of seismic waves in different geological structures are different. (Yu et al., 2021) calculated the daily ApNe value of the corrected strain from January 1,2011 to January 1,2014, constructed the threshold interval according to the ApNe mean and 2 times the standard deviation, and accumulated the results exceeding the threshold. Their experiments selected six stations : GZ, XM, ZT, YS, RH and TC. Among them, GZ station is the closest to the epicenter, XM station is the second from the epicenter, and ZT station is the third from the epicenter. The anomalous accumulation results of ApNe values of

surface strains at stations GZ, XM and ZT are given in the figure below, from which it can be seen that the number of accumulations at the more distant station ZT is more than that at station XM, and the number of accumulations at station XM is more than that at station GZ, and the fitting results of the anomalous accumulations of the three stations all show a strong S-shape anomaly. We believe that it may be due to the fact that the geology near the epicenter may be relatively hard or uniform, making it difficult for seismic precursor signals to be significantly transmitted or captured by stations. The geological conditions of stations far away may be more conducive to the propagation or amplification of signals, which makes it easier to receive abnormal signals.

[Figure]

Figure 6: Accumulation of ApNe anomalies at the GZ, XM, and ZT stations. The solid dots are the cumulative ApNe anomaly counts from 2011-2014. The black, pink, and blue lines are the sigmoidal fits at the GZ, XM, and ZT stations. The green line is the sigmoidal fit after the earthquake at the GZ station. The vertical red line is the day of the 2013 Lushan earthquake.

(2) Locality of earthquake precursors : Some earthquake precursors (such as gas release, electromagnetic anomalies, etc.) have strong locality, which may not be significant near the epicenter, but more obvious at specific locations in the periphery of the epicenter. Kumar et al., (2021) analyzed ionospheric anomalies associated with the 2019 Indonesia earthquake (Mw=7.4) using GPS and VLF measurements from multiple stations. They found that the disturbance observed at the place closest to the epicenter is the smallest, and they believe that the ionospheric disturbance induced by the earthquake depends not only on the distance between the observation and the epicenter, but also on the direction of the observation point relative to the epicenter. This may be due to local crustal fissures, stress concentration areas or other geological features.

(3) Non-linear propagation of signals: Seismic precursor signals may be affected by a variety of factors during propagation, such as wave scattering, attenuation, and reflections and refractions between different strata. These non-linear propagation phenomena may lead to a weakening or complication of the signal near the epicentre, thus making it difficult for close stations to receive it effectively.

(4) Effect of depth of source and type of earthquake: The depth of the source and the type of earthquake also affect the distribution of precursor signals. Precursor signals from deep earthquakes may be more difficult to capture at shallow stations near the epicentre, and different types of earthquakes (e.g., strike-slip, backlash, etc.) may also result in different propagation characteristics of the precursor signals.

So it is reasonable that this phenomenon occurs. In this paper, we pay more attention to the similarity of the anomalies received by each station for the same earthquake before the earthquake, and we will further explore the reasons for this phenomenon in the next study.

**References**

Chen, Y., Shu, T., Zhou, X. K., Zheng, X. Z., Kawai, A., Fueda, K., Yan, Z., Liang, W., and Wang, K. I. K.: Graph Attention Network With Spatial-Temporal Clustering for Traffic Flow Forecasting in Intelligent Transportation System, IEEE TRANSACTIONS ON INTELLIGENT TRANSPORTATION SYSTEMS, 24, 8727-8737, 10.1109/TITS.2022.3208952, 2023.

Chi, C., Li, C., Han, Y., Yu, Z., Li, X., and Zhang, D.: Pre-earthquake anomaly extraction from borehole strain data based on machine learning, Scientific Reports, 13, 10.1038/s41598-023-47387-z, 2023.

Kumar, S., Tripathi, G., Kumar, P., Singh, A. K., and Singh, A. K.: Ionospheric perturbations observed due to Indonesian Earthquake (Mw=7.4) using GPS and VLF measurements at multi-stations, ACTA GEODAETICA ET GEOPHYSICA, 56, 559-577, 10.1007/s40328-021-00345-5, 2021.

Nishimura, T.: Triggering of volcanic eruptions by large earthquakes, GEOPHYSICAL RESEARCH LETTERS, 44, 7750-7756, 10.1002/2017GL074579, 2017.

Qiu, Z., Yang, G., Tang, L., Guo, Y., and Zhang, B.: Abnormal Strain Changes Prior to the M7.0 Lushan Earthquake Observed by a Borehole Strainmeter at Guzan, Journal of Geodesy and Geodynamics, 35, 158-161+166, 10.14075/j.jgg.2015.01.036, 2015.

Yu, Z. N., Zhu, K. G., Hattori, K., Chi, C. Q., Fan, M. X., and He, X. D.: Borehole Strain Observations Based on a State-Space Model and ApNe Analysis Associated With the 2013 Lushan Earthquake, IEEE ACCESS, 9, 12167-12179, 10.1109/ACCESS.2021.3051614, 2021.

---

## Author Comment (AC2)

**Response to Reviewer:**
I am very grateful to your comments for the manuscript. Thank you for your advice. All your suggestions are very important. They have important guiding significance for our paper and our research work. We have revised the manuscript according to your comments. The response to each revision is listed as following:

**Comment 1**
While their manuscript provides a comprehensive overview of the techniques used, such as Variational Mode Decomposition (VMD) and Graph Wavenet Neural Network (GWN), it lacks a detailed justification for the choice of these specific methods over more traditional approaches. The manuscript would benefit from a clearer explanation of why these complex techniques were chosen and what specific advantages they offer when analysing borehole strain data, especially compared to simpler methods such as band filtering or traditional statistical models. For example, the use of VMD could be replaced by simpler signal processing methods, and the rationale for using a neural network for prediction, which could be achieved using conventional signal processing methods, is not convincingly presented.

**Response**:
        Thanks for your suggestion.
Borehole strain observation is the observation of crustal strain by installing strain sensors in boreholes, which has the advantages of high accuracy, wide bandwidth and strong anti-interference ability, and its observation data are widely used in the research of earthquake precursors and other aspects. Seismic signal is a typical non-stationary, non-linear time series data. Due to the complexity of the earth's structure, seismic signals are accompanied by various kinds of noise at the stages of generation, propagation and acquisition, and the band difference between the weak useful signals and the noise from the deep underground is very small and difficult to distinguish, and the extraction of effective microseismic signals from contaminated microseismic signals is a prerequisite for the subsequent analyses and researches, which will affect the final analysis of the whole seismic event.

There are many processing methods for seismic signals, including many common and effective methods. (Ma et al., 2011) used digital filtering techniques to study the body strain and barometric pressure data from Yixian station from 2002 to 2007, removed the long-period components in the raw data, and analysed the high-frequency spectral characteristics of the body strain with the fast Fourier transform. (Deng et al., 2015) used the Fourier transform to generate a spectral decomposition method for high-resolution seismic images based on information such as the frequency-amplitude spectrum of the signal, which was applied in the extraction of weak signals from deep reflection earthquakes. However, the Fourier transform is insufficient for non-smooth signals, and the Fourier transform-based filtering method for non-smooth signals will have problems such as signal distortion. (Zhang, 2018) used continuous wavelet

transform to analyse the time-frequency analysis of the strain data of the borehole at Guza, extracted the strain anomalies in the time-frequency spectra, and analysed the correlation between the strain anomalies and the anomalies of the seismic precursors. However, the wavelet transform has the problems of wavelet base selection, frequency domain overlap and threshold uncertainty, which is not suitable for analysing nonlinear smooth signals whose frequency varies with time, and is also not suitable for local analysis of signals. Unlike wavelet decomposition, EMD can represent the signal as an extension of basis functions which come directly from the signal itself, without defining the wavelet bases by itself, and the decomposition done is based on the intrinsic characteristics of the signal. The EMD method can smooth a non-smooth signal to obtain a series of components with different frequencies (IMFs), and by such a method a non-smooth, non-linear signal can be decomposition into smooth signals with different time scales (Lei et al., 2022). The Hilbert-Huang transform (HHT), which consists of EMD and Hibert transform, is an adaptive signal processing method that performs modal decomposition based on the characteristics of the data itself, and it has clear physical significance for the processing of nonlinear smooth signals. (Yang et al., 2014) used HHT to analyse the marginal spectral features of the unexplained large tensile jumps recorded in the borehole body strain at the Qianling seismic station in February-June 2012, and judged that the main cause of this strain anomaly was a power supply problem. However, EMD also has drawbacks, such as the presence of mode aliasing, endpoint effects, and difficulty in determining the stopping conditions. In order to overcome these drawbacks, Konstantin Dragomiretskiy and Dominique Zosso proposed VMD(Dragomiretskiy and Zosso, 2014). VMD is a theoretically well-founded technique and is more resistant to sampling and noise compared to EMD. Compared with the recursive decomposition mode of EMD, VMD turns the signal decomposition into a variational decomposition mode, which is in essence a set of multiple adaptive Wiener filters, VMD can achieve adaptive segmentation of each component in the signal frequency domain, and is able to effectively overcome the pattern aliasing phenomenon generated in EMD decomposition, with stronger noise robustness and weaker endpoint effects than EMD. Therefore, the VMD method is suitable for analysing nonlinear nonsmooth signals such as step, glitch and burr. The VMD method has been widely used in fields such as geosciences, and the results of processing seismic signals are significantly better than the other signal processing methods mentioned above(Zhang et al., 2022; Rao et al., 2024; Liu et al., 2016; Li et al., 2018).

Deep learning is a branch of machine learning, which is a machine learning algorithm based on neural networks. Unlike traditional time-series analysis methods, deep learning can introduce more external information and is not limited to extrapolating data based on historical trends and seasonality. Deep learning can automatically learn features and patterns from raw data, and is able to learn multiple layers of abstract features. By increasing the number of layers in the neural network, more complex and abstract features can be learnt, leading to more accurate classification and prediction. For the GWN graph neural network we used, the use of dilated causal convolutional layers avoids the limitation of needing a large number of layers to process long time

sequences, enabling the model to effectively capture the dependencies of long time sequences; through the adjacency matrix, it is able to learn the hidden spatial dependencies through node embeddings; in contrast to recurrent neural network (RNN)-based models, the GWN convolutional network architecture allows for parallel computation , which solves the gradient vanishing/exploding problem of RNN when dealing with long time sequences.

**Modification:**

A new paragraph was added to line 37 of the original manuscript: "There are many processing methods for seismic signals, including many common and effective methods. (Ma et al., 2011) used digital filtering techniques to study the body strain and barometric pressure data from Yixian station from 2002 to 2007, removed the long-period components in the raw data, and analysed the high-frequency spectral characteristics of the body strain with the fast Fourier transform. (Deng et al., 2015) used the Fourier transform to generate a spectral decomposition method for high-resolution seismic images based on the frequency-amplitude spectrum of the signal, which was applied in the extraction of weak signals from deep reflection earthquakes. (Zhang, 2018) used the continuous wavelet transform method to analyse the time-frequency analysis of the borehole strain data from Guza Station, extracted the strain anomalies in the time-frequency spectrum, and analysed the correlation between the strain anomalies and the seismic precursor anomalies. EMD method can smooth the non-smooth signals to obtain a series of components with different frequencies (IMF), by which the non-smooth, non-linear signals can be decomposed into smooth signals with different time scales (Lei et al., 2022). (Yang et al., 2014) used HHT to analyse the marginal spectral features of the unexplained large tensile jumps recorded in the borehole body strain at the Qianling seismic station in February-June 2012, and judged that the main cause of this strain anomaly was a power supply problem. However, EMD suffers from mode aliasing phenomenon, endpoint effect, and difficulty in determining the stopping condition. Compared with the recursive decomposition mode of EMD, VMD transforms the signal decomposition into a variational decomposition mode, which is essentially a set of multiple adaptive Wiener filters, and VMD can realise the adaptive segmentation of each component in the frequency domain of the signal, which can effectively overcome the mode aliasing phenomenon generated by EMD decomposition, and has a stronger noise robustness and a weaker end-point effect than EMD. Therefore, the VMD method is suitable for analysing nonlinear nonsmooth signals such as step, jumps and burr. The VMD method has been widely used in fields such as geosciences, and the results of processing seismic signals are significantly better than the other signal processing methods mentioned above (Zhang et al., 2022; Rao et al., 2024; Liu et al., 2016; Li et al., 2018)".

**Comment 2**

Furthermore, while you mention that VMD was used to pre-process the data by removing annual trends and tides, there is no explanation of how this pre-processing

specifically improves the performance of the GWN model. This gap makes it difficult to assess the necessity and effectiveness of VMD within your analysis pipeline. A more detailed description of the role and impact of VMD could help clarify its contribution to your results.

**Response**:

Thanks for your suggestion.

Our idea of using VMD to do data preprocessing is due to the presence of step, glitch and burr in the raw data, which are anomalous conditions due to the data monitoring process, and may override the real information of seismic signals that we need. (Chi et al., 2019) used VMD to do the processing of 1-month surface strain data. They decomposed Sa into five components and found that IMF1 is the trend term. They did the Fourier transform of IMF2 and found that the frequencies of the signal are mainly concentrated in $f_1=1.157 \times 10^{-5}$ Hz and $f_2=2.232 \times 10^{-5}$ Hz, which correspond to the semi-diurnal and diurnal wave frequencies of the Earth's tides, respectively. It is considered that IMF2 corresponds to the influence of the Earth's tides, and the IMF3-IMF5 components all contain a large amount of strain signals, and the remaining IMF3-IMF5 components are retained as the object of study. Thus the reason we chose VMD is not to improve the model performance, but only to remove the necessary influences.

**Comment 3**

In addition, while you report an increase in anomalous days 15-32 days prior to the earthquake, with a significant acceleration observed in the 20 days prior, the manuscript does not provide detailed statistical analyses or margins of error for these observations. Such information is crucial for understanding the robustness of your results. I suggest adding confidence intervals or error bars to make the reliability and statistical significance of your results clearer.

**Response**:

Thanks for your suggestion.

The VMD-GWN model we used uses a dual output based on upper and lower bounds at the output layer, and the output is a prediction interval which has nearly the same effect as the confidence intervals you mentioned, so our prediction results already come with a certain margin of error. However, as you said, Fig. 9 in the original manuscript does not indeed convey the reliability and statistical significance of the results in a completely clear way, so we have given the statistics of the anomaly rates. For the judgement condition of anomalous days in the original manuscript, it is only a judgement of whether each day is anomalous or not, and it is not clear the exact number of anomalies. Therefore, according to your suggestion, we made a count of the judgement results that met the conditions in each abnormal day, and took the statistical results as the number of abnormalities per day, and calculated the abnormal rate per day based on the number of abnormalities per day, and the statistical results of the abnormal rate per day are shown in Fig. 1 below.

[Figure]

**Figure 1. Daily anomaly rate statistics for four stations. Different coloured bars represent the daily anomaly rates of different stations. (a) Daily anomaly rate statistics from 62 days before to 3 days after the earthquake. (b) Daily anomaly rate statistics from 32 days before the earthquake to 15 days before the earthquake.**

As shown in Fig. 1(a), in the time range from 60 days before the earthquake to 33 days before the earthquake, all four stations showed only a very small number of anomalies until 32 days before the earthquake, when anomalies appeared at a number of stations, and the anomalies also increased significantly, with Haiyuan station showing the most significant number of anomalies. The dashed box in Fig. 1(a) corresponds to the time period from 32 days before the earthquake to 15 days before the earthquake, and the details are shown in Fig. 1(b). We find that there are several stations with anomalies at the same time in the 32, 21, 20, and 17 days before the earthquake, among which all four stations have anomalies on the 20th day before the earthquake, and from the value of the anomaly rate, the anomaly rate on the 17th day before the earthquake has a very obvious decrease, so we believe that a turning point occurs on the 20th day before the earthquake, which corresponds to the time when the accelerating effect of the S-shape fitting is the most obvious. After the 17th day before the earthquake, there was a quiet period where no station detected anomalies until the earthquake occurred.

The combination of Fig.1 and Fig.9 in the original manuscript can more fully present the process of abnormal changes before the earthquake.

**Modification:**

Add a new paragraph after line 302: "For the judgement condition of abnormal days in the original manuscript, it is only a judgement of whether each day is abnormal or not, and it is not clear the specific number of abnormalities. Therefore, we made a count of the judgement results that met the conditions in each abnormal day, and took

the statistical result as the number of abnormalities per day, and calculated the abnormal rate per day based on the number of abnormalities per day, and the statistical result of the abnormal rate per day is shown in Fig. 10 below.

[Figure]

Figure 10. Daily anomaly rate statistics for four stations. Different coloured bars represent the daily anomaly rates of different stations (a) Daily anomaly rate statistics from 62 days before to 3 days after the earthquake. (b) Daily anomaly rate statistics from 32 days before the earthquake to 15 days before the earthquake.

As shown in Fig. 10(a), in the time range from 60 days before the earthquake to 33 days before the earthquake, all four stations showed only a very small number of anomalies until 32 days before the earthquake, when anomalies appeared at a number of stations, and the anomalies also increased significantly, with Haiyuan station showing the most significant number of anomalies. The dashed box in Fig. 10(a) corresponds to the time period from 32 days before the earthquake to 15 days before the earthquake, and the details are shown in Fig. 10(b). We find that there are several stations with anomalies at the same time in the 32, 21, 20, and 17 days before the earthquake, among which all four stations have anomalies on the 20th day before the earthquake, and from the value of the anomaly rate, the anomaly rate on the 17th day before the earthquake has a very obvious decrease, so we believe that a turning point occurs on the 20th day before the earthquake, which corresponds to the time when the accelerating effect of the S-shape fitting is the most obvious. After the 17th day before the earthquake, there was a quiet period where no station detected anomalies until the earthquake occurred. The combined analysis of Fig. 9 and Fig. 10 gives a fuller picture of the process of pre-seismic anomalous changes."

**Comment 4**

I also noticed that you observed an increase in anomalous days one to three days after the earthquake and attributed this to aftershocks. While this observation is interesting, it seems to have little to do with the main focus of your study on earthquake prediction. It would be helpful to clarify how this post-seismic analysis relates to the main goal of earthquake prediction and to discuss its significance in the context of your overall results.

**Response**:

Thanks for your suggestion.
You are right to raise the point that the focus of our research is really not on post-earthquake analysis, so we have removed that part to ensure consistency of thought in the article.

**Modification:**

Lines 308-314 in the original manuscript see were deleted: "(2) Anomalous days were also observed at all four stations on August 9 and 10 after the earthquake. Zhong et al., (2020) studied the IR anomalies and ionospheric anomalies in the same area before the Jiuzhaigou earthquake and found that the thermal radiation continued to increase until August 14, and the ionospheric anomalies were detected on August 11 and 15 after the earthquake. Xu et al., (2021) studied the ionospheric TEC anomalies of the Jiuzhaigou earthquake, and the anomalies were detected four days after the earthquake using different methods. The results of our study are consistent with current research, and anomalies were observed for several days after the earthquake. We believe that the post-earthquake anomalies were more likely due to the frequent occurrence of post-earthquake aftershocks.".

**Comment 5**

In addition, the use of an S-shaped function to fit the cumulative results of anomalous days is mentioned, but the manuscript does not adequately explain why this particular fitting method was chosen or how it compares to other models. A more detailed discussion of this choice and the associated findings would improve the reader's understanding of your analytical approach.

**Response**:

Thanks for your suggestion.
(Bufe and Varnes, 1993) and (Bufe et al., 1994) found that the clustering of intermediate events prior to a large shock leads to a regional increase in the cumulative Benioff strain $\varepsilon(t)$, which can be fitted by a power-law time-destruction relationship:

$$\varepsilon(t) = A + B(t_f - t)^m \tag{1}$$

where $A$ and $B$ are constants, $0 < m < 1$ is a constant for adjusting the power law, and $t_f$ is the predicted mainshock time, i.e., the critical point in time at which the process of accelerating cumulative Benioff strain (cumulative energy) takes place. This behaviour has been interpreted as a critical process prior to the movement of a

large earthquake towards a critical point (i.e., the mainshock). (Bufe and Varnes, 1993) justified Equation 1 using a simple damage mechanics model.

(De Santis, 2014) studied the 2009 L'Aquila and 2012 Emilia earthquakes based on earthquake catalogues. Equation 2 is an inverse diffusion equation for the spatial proximity of seismic events to the epicentre and is used to fit the distribution of seismic distances over time within 200 km of the epicentre region:

$$r(t) = D_r \cdot (t_f - t)^{m_2}$$

(2)

where $D_r$ and $m_2$ are constants and tf represents the critical point in space at which seismic events are focused. Equation 3 fits the distribution of the time interval between seismic events $\tau(t)$ over time:

$$\tau(t) = D_\tau \cdot (t_f - t)^{m_3} \tag{3}$$

where $D_\tau$ and $m_3$ are constants and tf represents the critical point in time at which seismic events are focused.

Equation 2 and Equation 3 specifically show the manifestation of this energy accumulation in space and time, and this process in time and space is known as the spatio-temporal focussing phenomenon before the mainshock. The research idea of this paper is the extraction of multi-station pre-seismic anomalies based on spatio-temporal features, and the fitting method proposed above has good results in spatio-temporal and this fitting method has theoretical support and physical significance, so for the anomalous results in our original manuscript, the fitting is done by using the S-type function. De Santis et al., (2017) used Swarm magnetosatellite data to study the 2015 Nepal earthquake and proposed an S-shaped fitting function in anomalous cumulative analysis; they found that S-shaped fitting was significantly superior to linear fitting.

**Modification:**

Add a new paragraph on line 276: "(Bufe and Varnes, 1993) and (Bufe et al., 1994) found that the clustering of intermediate events prior to a large shock leads to a regional increase in the cumulative Benioff strain $\varepsilon(t)$, which can be fitted by a power-law time-destruction relationship:

$$\varepsilon(t) = A + B(t_f - t)^m \tag{10}$$

where $A$ and $B$ are constants, $0 < m < 1$ is a constant for adjusting the power law, $t_f$ is the predicted time of the mainshock, i.e., the critical point in time for the acceleration process of the cumulative Benioff strain (cumulative energy). This behaviour has been interpreted as a critical process preceding the movement of a large earthquake towards a critical point (i.e., the mainshock). (Bufe and Varnes, 1993) justify Equation 1 with a simple model of damage mechanics. (De Santis, 2014) studied the 2009 L'Aquila and 2012 Emilia earthquakes based on seismic catalogues, showing concretely how this accumulation of energy in space and time manifestations. The research idea of this paper is the extraction of multi-station pre-earthquake

anomalies based on spatio-temporal features, and the fitting method proposed above has good results in spatio-temporal and this fitting method has theoretical support and physical significance, so for the anomalous results in our original manuscript, we use the S-type function to do the fitting."

**Comment 6**

Finally, the manuscript suggests that the pre-earthquake anomalies are due to strain energy diffusion near the epicentre. This claim appears to have been made without a solid empirical or theoretical basis in the text. It would be beneficial if you could provide additional evidence or references to support this assumption or discuss alternative explanations for the observed anomalies.

**Response**:

Thanks for your suggestion.

Add a new paragraph before line 302:"(Zhang et al., 2018) analysed the temporal and spatial evolution characteristics of the precursor anomalies of the Jiuzhaigou earthquake, and found that the short-term phases of the precursor anomalies of the Jiuzhaigou earthquake are divided into two phases, $\gamma_1$ and $\gamma_2$, among which the anomalies in the $\gamma_2$ phase are deformation anomalies, which are manifested as the expansion of anomalies from the near-source area to the outside of the epicentre. (Wang et al., 1984) found that the extension of precursor anomalies to the periphery of the epicentre was due to the subcritical extension of cracks, and justified their conclusions based on the inversion results of the precursor observations of resistivity. (Guo et al., 2020) analysed the deformation process in the unstable state of a fault and defined the meta-stable (or sub-stable) state of a fault as the transition stage from peak stress to fast destabilising critical stress throughout the slow loading and fast unloading. At this stage, the accumulated strain energy starts to be released.".

**Comment 7**

The manuscript is generally well written, but there are some areas where the English could be improved (e.g., "This unique geographic location makes earthquakes a common occurrence"). Also, some typographical errors need to be corrected (e.g., "sevesral" and "pro-seismic"). The figures, especially Figures 3, 4 and 7, are too small and difficult to read, which makes them difficult to understand.

I hope that these comments will be helpful in revising your manuscript. Clarifying these points will not only strengthen the scientific rigour of your study, but will also make your results more accessible and meaningful to the research community.

Best regards.

**Response**:

Thanks for your suggestion.

Line 59-61. Modified "The Sichuan Basin is located at the junction of the

Asia-Europe Plate and the Indian Ocean Plate, and is influenced by neighboring mountain ranges and plateaus, forming sevesral fracture zones. This unique geographic location makes earthquakes a frequent event (Zhang, 2023). " It is modified to "The Sichuan Basin is at the junction of the Asia-Europe Plate and the Indian Ocean Plate, and is influenced by the neighbouring mountain ranges and plateaus, forming several fracture zones, and its unique geographic location has led to frequent earthquakes within Sichuan(Zhang, 2023). ".

Line 60. Modified "sevesral". It is modified to "several".

Line 16. Modified "pro-seismic". It is modified to "pre-seismic".

Modifications were made to Figures 3, 4 and 7 in the original manuscript. The results of the modifications are shown below:

[Figure]

**Figure 3: Data sets of $S_a$ components for Linxia, Guza, Haiyuan, and Gaotai stations. (a) $S_a$ component data of each station for training dataset; (b) $S_a$ component data of each station for test dataset. Red dotted line indicates time of Jiuzhaigou earthquake.**

[Figure]

**Figure 4: Plot of decomposition results of $S_a$ data using VMD method at Linxia station.**

[Figure]

| index | linxia | guza | haiyuan | gaotai |
|---|---|---|---|---|
| 1 | -0.94099 | -5.22487 | -1.50890 | 0.10352 |
| 2 | -0.24329 | -4.98184 | 0.28445 | -0.97876 |
| ⋮ | ⋮ | ⋮ | ⋮ | ⋮ |
| 60 | -0.14255 | 1.39405 | -0.11844 | 0.25033 |
| 61 | -0.24920 | 1.40841 | 0.33341 | -0.11321 |
| 62 | 0.03218 | 1.67455 | 0.08946 | 0.13671 |
| ⋮ | ⋮ | ⋮ | ⋮ | ⋮ |
| 120 | 0.33519 | -0.04530 | 0.49508 | -0.19615 |
| 121 | -0.05818 | -0.00030 | 0.07047 | -0.63347 |
| ⋮ | ⋮ | ⋮ | ⋮ | ⋮ |
| 525480 | -0.42070 | -0.09572 | 0.10945 | 0.32081 |
| 525481 | -0.38017 | -0.26278 | -0.15934 | 0.02006 |
| ⋮ | ⋮ | ⋮ | ⋮ | ⋮ |
| 525539 | -0.68837 | -0.71587 | 0.07619 | -0.00601 |
| 525540 | -0.97372 | -0.66513 | 0.32384 | 0.47323 |
| 525541 | -1.09317 | -0.76759 | 0.45178 | -0.02135 |
| ⋮ | ⋮ | ⋮ | ⋮ | ⋮ |
| 525599 | 0.33423 | 2.13201 | -0.14036 | -0.33949 |
| 525600 | 0.05900 | 2.27773 | -1.61969 | 0.29316 |

**(a)** Total number of samples is 525481; total number of labels is 525481 training set: validation set = 3:1

[revised manuscript text omitted]